# Distinguishing the Effects of Stress Intensity and Stress Duration in Plant Responses to Salinity

**DOI:** 10.3390/plants12132522

**Published:** 2023-07-01

**Authors:** Caitlin DiCara, Keryn Gedan

**Affiliations:** Department of Biological Sciences, Columbian College of Arts & Sciences, The George Washington University, Washington, DC 20052, USA; cdicara17@gmail.com

**Keywords:** salt water intrusion, salinity stress, plant response to stress

## Abstract

Species-specific variation in response to stress is a key driver of ecological patterns. As climate change alters stress regimes, coastal plants are experiencing intensifying salinity stress due to sea-level rise and more intense storms. This study investigates the variation in species’ responses to presses and pulses of salinity stress in five glycophytic and five halophytic species to determine whether salinity intensity, duration, or their interaction best explain patterns of survival and performance. In salinity stress exposure experiments, we manipulated the intensity and duration of salinity exposure to challenge species’ expected salinity tolerances. Salinity intensity best explained patterns of survival in glycophytic species, while the interaction between intensity and duration was a better predictor of survival in halophytic species. The interaction between intensity and duration also best explained biomass and chlorophyll production for all tested species. There was interspecific variability in the magnitude of the interactive effect of salinity intensity and duration, with some glycophytic species (*Persicaria maculosa*, *Sorghum bicolor*, and *Glycine max*) having a more pronounced, negative biomass response. For the majority of species, prolonged stress duration exacerbated the negative effect of salinity intensity on biomass. We also observed an unexpected, compensatory response in chlorophyll production in two species, *Phragmites australis* and *Kosteletzkya virginica*, for which the effect of salinity intensity on chlorophyll became more positive with increasing duration. We found the regression coefficient of salinity intensity versus biomass at the highest stress duration, i.e., as a press stressor, to be a useful indicator of salinity tolerance, for which species’ salinity-tolerance levels matched those in the literature. In conclusion, by measuring species-specific responses to stress exposure, we were able to visualize the independent and interactive effects of two components of a salinity stress regime, intensity, and duration, to reveal how species’ responses vary in magnitude and by tolerance class.

## 1. Introduction

Species-specific variations in response to stress fundamentally underlie community composition and ecosystem services; quantifying this interspecific variation can help forecast future distributions and predict community dynamics. For example, interspecific variation in stress tolerance drives succession [1,2], spatial patterning such as zonation [3], ecological filters [4], species interactions [5,6], and species distributions [7,8]. Stress events can be broadly categorized as press stressors (i.e., long-term changes in an environmental variable) and pulse stressors (i.e., shorter, punctuated events) [9,10].

Salinity is a near universal stressor for terrestrial plants. Elevated salt concentrations induce osmotic stress, mimicking drought conditions and making it more difficult for plants to uptake the water necessary to perform vital functions, including photosynthesis and maintaining turgor pressure [11]. Additionally, salt can have toxic effects on plant tissues [12]; it can reduce germination, nutrient uptake, seed production or crop yield, and growth, and can ultimately cause senescence and plant death [12,13,14].

Salinity is often dynamic in the soil environment and can occur as a press or pulse stressor [15]. Sea-level rise (SLR) is a press stressor that acts over longer timescales, as salinity steadily rises when tidal waters permeate the soil and groundwater [10]. This primarily impacts stress intensity, as salt concentrations gradually rise over decadal timescales. Saltwater intrusion (SWI; defined as the movement of saltwater into inland areas via groundwater) and storm events act as pulse stressors, with seawater overtopping soil during acute flooding events and causing punctuated rises in groundwater and porewater salinity at local and regional scales [16]. The frequency and strength of storm and SWI events are anticipated to increase due to elevated sea surface temperatures in tandem with gradually rising sea levels [17]. It is therefore imperative to study the interaction between salinity intensity and stress duration in order to anticipate its impact on coastal plant communities in the face of pulse and press stressors [17,18]. This is the impetus for our study, which seeks to investigate, on a species-level, how plants with different salinity adaptations will respond to elevated stress along multiple axes of exposure.

Controlled greenhouse experiments and field observations have demonstrated that plant ecophysiological responses to increased salinity are species-specific, with marked differences in how species tolerate and recover from seawater inundation events [10,18,19,20]. Plant species can also vary in thresholds of tolerance based on the duration of salinity exposure [10]. Some non-halophytic plant species can recover after short-term exposure to elevated salinity levels as long as post-pulse conditions allow for a recovery period, i.e., lowered interstitial salinities [21,22]. This is a source of resilience for coastal plant communities experiencing pulses of salinity. Yet, if elevated salinity persists over time and becomes a press stressor, or the post-pulse environment is not suitable for plant recovery, salt-sensitive species are filtered out of the community assemblage [10,21]. Interspecific variation in tolerance and recovery patterns between species can result in a shift in community composition. This, in turn, can alter ecosystem function [23,24,25]. Understanding which plant species are tolerant to different combinations of salinity intensity and duration can offer insight into how communities are likely to shift in composition, and therefore function, over time with SLR and SWI [24,25].

To investigate how pulses and presses of salinity stress affect the survival and fitness of coastal plant species, we manipulated two variables of salinity exposure that represent the pulse and press stressors occurring in coastal ecosystems: (1) intensity, or the concentration of salt in water, measured in parts per thousand (PPT); and (2) duration, or the length of time that plants were exposed to salt water, measured in days per month (DPM) [18]. We also calculate a novel stress index that integrates salinity stress intensity and duration, which we call the salt stress exposure (SSE) index.

The SSE index was inspired by the growing degree days index, which has intuitive units and is extremely useful in relating temperature and organismal physiology. The growing degree days index tabulates the time when plant growth can occur by summing the number of degrees over a baseline temperature which accumulate each day over a growing season, and the summation of which describes the total growing time that falls within benign operating conditions [26]. Every degree over the threshold temperature contributes to the growth of the plant, and growing degree days are highly correlated with plant production and yield. Building on this framework of growing degree days, we estimate the SSE index by multiplying the intensity treatment by the duration treatment to accumulate salinity exposure “stress-time”. SSE combines intensity and duration into a single variable, as a simple representation of the interaction between intensity and duration.

As species vary greatly in their ability to tolerate salinity, this study aimed to test salinity tolerance in a group of coastal and upland species that exhibit a diverse array of adaptations to salinity [27] (Table 1). Species fall into two broad tolerance groups: halophytes and glycophytes. Halophytic plants have evolved specialized adaptations to cope with saline conditions, such as synthesizing organic solutes and accumulating stress metabolites such as proline, calcium, or potassium ions to maintain osmotic balance between internal and external environments [12,28]. Additional adaptations for salt tolerance include low rates of sodium and chlorine ion transport to the leaves and compartmentalization in vacuoles or cell walls to prevent toxic buildup in the cytoplasm [11]. A select number of halophytes are also able to exude salt from salt glands in their leaves to prevent toxicity [29,30]. Having one or more of these adaptations allows halophytic species to avoid toxicity from an excessive buildup of salt ions. Glycophytes lack these adaptations and are therefore more susceptible to salinity stress.

Coastal and tidal wetland species are halophytic and adapted to repeated inundation events and elevated levels of salinity. We expected halophytic species (Table 1) to maintain biomass and chlorophyll production at elevated levels of salinity (both intensity and duration). Upland plant species tested within this study included native and introduced herbs as well as several common crop species (Table 1). We expected to see less salinity tolerance in upland, glycophytic species, resulting in lower biomass and lower chlorophyll production in response to increasing salinity intensity and duration.

**Table 1 plants-12-02522-t001:** Known halophytic adaptations of the focal species in this study.

Species	Tolerance Class	Adaptations to Saline Conditions
*Kosteletzkya virginica*	higher tolerance, native	Accumulates inorganic compounds and synthesizes organic compounds to maintain osmotic balance [31], including proline [31,32]. Mucilage in shoot, stems, and roots assists in regulating water ascent and ion transport [33].
*Panicum amarum*	higher tolerance, native	Adapted to dune systems, where plants are subjected to salt spray, storm surges, inundation, and other stresses; can withstand extended drought [34].
*Panicum virgatum*	higher tolerance, native	Salt excretion glands [35]; synthesizes organic compounds (e.g., proline) to maintain osmotic balance.
*Paspalum floridanum*	higher tolerance, native	Observed in saline ditches adjacent to farm fields (pers. comm., Chris Miller).
*Phragmites australis*	higher tolerance, invasive	Salt excretion glands [36]; organic and inorganic compounds raise osmotic pressure [27]; oxidoreductase activity and glutathione metabolism [37]; free amino acids and sugars as osmolytes (namely proline and glutamine) [38]; ions (K^+^, Na^+^, Ca^2+^, Mg^2+^ and Cl^−^) in bundle sheath and mesophyll cells [39].
*Brassica napus*	lower tolerance, native	Accumulates proteins associated with protein metabolism and damage repair in saline conditions [40].
*Glycine max*	lower tolerance, crop, a genetically modified, chloride-excluding variety	Root length unaffected and root water content increased in mild to moderate salinity (3 to 12 PPT) [41].
*Persicaria punctata*	lower tolerance, native	Found in low-salinity (~5 PPT) tidal marshes [42].
*Persicaria maculosa*	lower tolerance, invasive	No published salt-tolerance information.
*Sorghum bicolor*	lower tolerance, crop	Acclimation at low salinity allowed internal regulation of Na^+^ and Cl^−^ concentrations during extended exposure to higher salt concentrations [43].

By developing physiological response curves based on exposure to various levels of salinity intensity and duration, this study investigated which metric(s) of salinity exposure explain variation in survival and plant ecophysiology (biomass and chlorophyll production) of halophytic and glycophytic species. In addition, we test the ability of a simplified salinity stress metric, the SSE index, to describe species tolerance to salinity stress, and compare this index to the alternative of models that include separate predictor variables of intensity and duration. The research significance of this study is to identify the metric of salinity that best explains plant physiology and mortality, detect whether this metric differs between glycophytes and halophytes, and identify, from amongst the tested crops, those which perform best in elevated salinity.

## 2. Results

### 2.1. Survival

Plant tolerance classes exhibited different responses to salinity treatments in survival. The glycophyte and halophyte groups had different models of best fit for the response variable of survival. In the glycophyte experiment, in which species were exposed to salinities ranging from 0 to 6 PPT, the model of best fit had salinity intensity alone as a fixed predictor (Table 2). In the halophyte experiment, in which species were exposed to salinities ranging from 0 to 24 PPT, the model of best fit for survival was the model with intensity and duration and their interaction as fixed predictors (Table 2); intensity and duration had independent and interactive effects on halophyte survival.

When the model of best fit was run for each species, intensity failed to significantly explain patterns in survival in any of the glycophytes, despite being the model of best fit. However, intensity significantly explained a decline in survival for the halophyte *K. virginica* (*p* = 0.00314) within the full model of intensity, duration, and their interaction. While most species exhibited high survival across salinity stress treatments, *K. virginica* experienced ≥50% mortality across all durations of 24 PPT. Additionally, *P. virgatum* experienced high mortality in salinity intensity treatments of 24 PPT; all individuals in the 30 DPM treatment of 24 PPT died prior to harvest. The remaining species (glycophytes: *P. punctata*, *P. maculosa*, *B. napus*, *S. bicolor*, *G. max*; halophytes *P. floridanum*, *P. amarum* and *P. australis*), maintained high levels of survival across nearly all treatments of intensity and duration (Table 3 and Table 4).

### 2.2. Biomass

Unlike survival, the same model provided the best fit to patterns in plant ecophysiology in the glycophyte and halophyte experiments. The model that included salinity intensity, duration, and their interaction best explained variation in standardized total biomass, standardized aboveground biomass, and chlorophyll (Table 2, Table 5 and Table 6).

Total biomass of native wetland halophyte *K. virginica* was negatively affected by salinity intensity to a similar degree across all levels of duration but produced greater biomass at low-intensity salinity (6 PPT) than in the freshwater control, indicating a nonlinear effect of salinity on total plant production. ANOVA analysis found significant differences in total biomass of *K. virginica* by intensity (F_3,91_ = 12.95, *p* < 0.001) (Figure 1).

When the model of best fit was run individually for each species to test for significant coefficients, intensity and duration had a significant, interactive effect on total biomass in 6 of 10 species: glycophytes *P. maculosa*, *G. max*, and *S. bicolor*, and halophytes *P. amarum*, *P. floridanum*, and *K. virginica* (Table 5). For all species except the halophyte *K. virginica*, the effect of PPT on biomass became more negative with increasing duration (Figure 2). In other words, prolonged salinity duration magnified the negative effect of salinity intensity on biomass. This effect was most pronounced in the glycophytic species *P. maculosa*, *S. bicolor*, and *G. max* (Figure 2). For the remaining four species, halophytes *P. virgatum* and *P. australis* and glycophytes *B. napus* and *P. punctata*, each produced the same biomass in all salinity stress treatments, regardless of intensity or duration. With more surviving replicates, there might have been a main effect of salinity intensity on *P. virgatum* in the halophyte experiment (Figure 2); however, the sample size was greatly reduced due to high mortality (Table 3).

#### Ranking Species by Salt Sensitivity Scores

Based on their performance in the highest stress intensity by duration combination, all the glycophytic species, with the exception of *P. punctata*, lost more biomass per unit increase PPT than any of the halophytic species tested (Table 7). From least to most salt-sensitive, the tested species ranked as follows (h: halophyte, g: glycophyte, c: crop): *P. punctata* (g), *P. australis* (h), *P. amarum* (h), *P. virgatum* (h), *K. virginica* (h), *P. floridanum* (g), *B. napus* (g, c), *S. bicolor* (g, c), *P. maculosa* (g), and *G. max* (g, c) (Table 7).

Based on this metric of salt sensitivity, the crop species *B. napus* and *S. bicolor* had similar magnitudes of biomass change per unit PPT, approximately −0.1 g/PPT, while the third crop tested, *G. max*, was the least tolerant of those tested, alongside the introduced glycophyte *P. maculosa*, with a coefficient >−0.5 g/PPT (Table 7). Additionally, based on this metric, the glycophyte *P. punctata* presented an anomaly in that this supposedly salt-sensitive species exhibited a non-negative effect of intensity at 30 DPM on total biomass.

Including *P. virgatum* in both the glycophyte and halophyte experiments allowed comparisons across the two. *P. virgatum* experienced a stronger negative effect of PPT on biomass during the halophyte experiment than the glycophyte experiment (top row, Figure 2). This was likely a function of the low maximum-salinity intensity (6 PPT) in the first experiment, which did not produce as strong an effect on *P. virgatum* biomass. Survival of *P. virgatum* was lower in the halophyte experiment than in the glycophyte experiment (Table 3 and Table 4).

### 2.3. Chlorophyll

Chlorophyll measurements were only collected in the halophyte experiment. Similar to biomass, the response of chlorophyll to the individual and interactive effects of salinity intensity and duration was species-specific, but three of the five species tested (natives *K. virginica* and *P. floridanum*, and invasive *P. australis*) exhibited a significant interaction between PPT and DPM. For both *K. virginica* and *P. australis*, the relationship between salinity intensity (PPT) and chlorophyll became more positive with increasing DPM. In other words, as duration increased, the effect of intensity changed from negative at 5 DPM (more intense stress reduced chlorophyll concentration) to positive at 30 DPM (more intense stress increased chlorophyll concentration). For *K. virginica* and *P. australis*, the highest concentrations of chlorophyll occurred in individuals grown in the most stressful combination of intensity and duration. The native wetland species *P. floridanum* exhibited a different type of interactive effect. For *P. floridanum*, the effect of intensity on chlorophyll became less positive with increasing duration (Figure 3). Additionally, and unlike *K. virginica* and *P. australis*, chlorophyll concentrations in *P. floridanum* were always lower at higher salinity intensities, across all levels of duration.

In *P. amarum*, chlorophyll concentration was significantly explained by DPM alone (*p* = 0.0133), with shorter duration salinity exhibiting lower levels of chlorophyll across all levels of intensity. However, *P. virgatum* exhibited little variation in chlorophyll concentration across any treatment level of intensity or duration, or their interaction.

## 3. Discussion

### 3.1. SSE Index Performance

In both the glycophyte and halophyte experiments, the SSE index failed to outperform other models of salinity stress in explaining effects on fitness and survival. The development of indices that can more simply describe salinity stress effects on plants and can generalize these effects to large groups of species, such as glycophytes or halophytes, would fulfill a research and communication need. Unfortunately, we found that the SSE was too simplistic of a metric to adequately describe salinity stress effects for either group of species. Rather, the independent and interactive effects of salinity intensity and duration are important in explaining patterns in plant responses to stress, and these effects were species-specific.

### 3.2. Salinity Stress Effects Varied Based on Performance Metric and Tolerance Class

The model of best fit for explaining patterns in survival varied between our two salinity tolerance classes. The intensity-alone model (PPT) was the best fit for survival of glycophytes. Patterns in survival for more susceptible species (i.e., those with no known salt-tolerance adaptations) were best explained by the salt concentration of exposure, rather than the duration of exposure. Moreover, the glycophyte experiment concluded after 60 days, which meant that lethal effects were relatively quick. There may be unobserved consequences of salinity stress on survival, for either group, that extend beyond our study period.

While intensity alone best explained patterns of survival among glycophytes, the independent and interactive effects between intensity and duration best explained survival in halophytic species. The same model of best fit, which included the main and interactive effects of intensity and duration, also explained the response of biomass in glycophytes and halophytes.

### 3.3. The Effects of Stress on Plant Performance

Survival to reproduction is a minimum prerequisite for plant fitness [44]. Identifying the component of a stress regime that imposes the most severe environmental filter on the plant community could help to inform important management implications when choosing mitigation strategies for coastal plant communities and agriculture [45]. As intensity had the most explanatory power in describing patterns in the survival of native (*P. punctata*), invasive (*P. maculosa*), and crop (*B. napus*, *G. max* and *S. bicolor*) glycophytic species, changes in the salinity concentration of soil porewater and groundwater are likely to have strong effects on plant communities, regardless of their duration.

Differences in salinity stress tolerance among species can affect community structure and function. While our study investigated individual species responses, Li and Pennings [46] investigated community dynamics in a factorial greenhouse mesocosm experiment exposing communities of six common tidal freshwater wetland plant species to three salinity intensity levels (3, 5, and 10 PPT) and five duration levels (5, 10, 15, 20, and 30 days per month) and found that salt-sensitive species were suppressed with increasing salinity exposure as both intensity and duration increased, resulting in shifts in species composition and decreased species richness. While these mesocosm communities were given 10 months in freshwater to recover, communities exposed to high salinity and duration were unable to recover species richness due to the loss of salt-sensitive species, highlighting the importance of survival for community development [46]. Based on our findings, we expect the less-tolerant glycophytes, such as *P. maculosa*, to be filtered from coastal plant communities first as coastal salinity regimes shift, while more tolerant species persist.

The effect of intensity on biomass became more negative with increasing duration for the majority of species (glycophytes: invasive *P. maculosa*, crops *S. bicolor* and *G. max*; halophytes: *P. amarum*, *P. floridanum*, and *K. virginica*). Biomass represents the total sum of a plants’ production throughout its lifespan [47], and this result indicates that a “generic” species response to salinity stress is a biomass cost to salinity intensity that is more consequential in longer exposures.

We also identified a compensatory physiological response to salinity exposure among some halophytes in the interaction between salinity intensity and duration and the response variable of chlorophyll. For two of the five halophytic species (invasive *P. australis* and native *K. virginica*), chlorophyll exhibited an increasingly positive relationship with intensity as duration increased. This is best explained by differences in the allocation of resources by plants, particularly those with halophytic adaptations, in response to stress, as plants are able to allocate and invest energy into chlorophyll and light harvesting as they develop. While chlorophyll concentrations most commonly decline with increasing salinity [48,49], salinity stress can prompt a compensatory response in some species, in which individuals invest assimilated carbon into cell maintenance and organelle construction rather than somatic growth; this results in reductions in biomass even while chlorophyll concentrations in the remaining tissues remain constant or increase [50]. Compensation in photosynthetic activity with increased stress was observed in response to other stressors, such as flooding [51]. Li et al. [52] found no significant reductions in net photosynthetic rate or chlorophyll with long-term flooding (60 days) in *P. australis* or the wetland grass species *Hemarthria altissima*. These results suggest that certain plant species, including the persistent invasive *P. australis*, can respond to both short- and long-term stress events by maintaining or increasing photosynthetic rates.

### 3.4. Species-Specific Plant Tolerance to Salinity

Given the range of halophytic adaptations found in the species included in this study, physiological response curves were expected to vary between species and salinity tolerance classes (Table 1). Salt-sensitivity analysis generally confirmed the expected tolerance levels of the tested species, with halophytes outperforming glycophytes, but it also yielded surprises (Table 5). For example, the comparison of the two glycophyte species in the genus *Persicaria* presented revealing differences. While we found *P. maculosa* to be one of the most salt-sensitive species included in the study, *P. punctata*, was surprisingly identified as the most salt-tolerant. There are few studies in the literature on the specific mechanisms of salt tolerance in *P. punctata*, but Humphreys et al. [42] noted this species in tidal marsh inventories dating back to 1980 in Virginia. Our study suggests that the species *P. punctata*, or at least the source population of *P. punctata* sampled for this study, has saline adaptations that would allow it to persist in coastal assemblages under increasing salinity.

We found the wetland species *P. australis* (an invasive) and *K. virginica* (a native) to be among the most tolerant to salinity stress of the species tested. *P. australis* is regarded to have broad salinity tolerance [53], and in this experiment it was able to maintain biomass production at all levels of salinity intensity and duration tested. This is likely due to this species’ salinity adaptations (Table 1). Our findings, coupled with previous studies, further explain the success of *P. australis* across a large range of salinities in coastal wetlands and its ability to outperform and dominate many native plant species in terms of salt tolerance [54]. *K. virginica*, one such native wetland species, was previously identified as a candidate for cultivation in salt-affected coastal areas [55]. In identifying *K. virginica* as a species of high-salinity stress tolerance in both pulse and press events, our study supports this application. We also found evidence for non-linear effects of salinity on plant biomass in *K. virginica*, where the mean total biomass peaked at 6 PPT, then significantly decreased at higher salinities (24 PPT). This apparent stimulation of biomass production at low levels of salinity further supports the use of *K. virginica* for restoration projects in fields experiencing elevated salinity.

For most of the species tested, the relationship between biomass and intensity became more negative with increasing duration. An acclimation period, such as would occur in an incremental or ramped stress event, could have changed this response, particularly for glycophytic species. For example, in a previous study, the crop species *S. bicolor* was able to acclimate to salinity in a pre-treatment period followed by a high-intensity salinity exposure [43]. Pre-treatment of salinity provides a ripe area for the future research and development of agricultural practices in coastal farmland.

The soybean *G. max* was the most sensitive species to salt stress of those included in this study, based on the strong, negative effect of salinity intensity and duration on biomass. *G. max* was previously demonstrated to decrease aboveground biomass, as well as leaf area, when exposed to salinity; this has been suggested as an adaptation to salt exposure by limiting transpiration and improving the plants’ hydration [41]. Despite prior demonstrations of salt-tolerance adaptations within this species to contain salt ions in roots and stem tissue [56], and despite the election of a chloride-excluding variety of *G. max*, the species did not perform well in this experiment, nor in field trials using the same variety in salt-stressed fields [57]. Taken together, we recommend *S. bicolor* as a better crop selection for salt-affected coastal areas than *G. max*. Our findings support the cultivation of *S. bicolor* in salt-affected areas, as encouraged by previous studies [58].

## 4. Materials and Methods

### 4.1. Species Selection and Sourcing

We selected five glycophytes, *Brassica napus*, *Glycine max*, *Persicaria maculosa*, *Persicaria punctata*, and *Sorghum bicolor*, and five halophytes, *Kosteletzkya virginica*, *Panicum amarum*, *Panicum virgatum*, *Paspalum floridanum*, and *Phragmites australis*, for salinity stress exposure experiments. These species were selected based on their importance in coastal and agro-ecosystems in the region, as well as seed availability. *P. australis* is a pervasive invasive species that is dominant in many coastal wetland systems in the eastern United States; all other halophytes were native species in the Mid-Atlantic region. Among the glycophytes, *P. maculosa* is an introduced species, and *B. napus*, *G. max*, and *S. bicolor* are important crops in the region. We intentionally included two sets of congeners, the two *Persicaria* species in the glycophyte experiment and the two *Panicum* species in the halophyte experiment. Additionally, the selected study species encompass a variety of ecological niches, habitat preferences, and known halophytic adaptations (Table 1). Therefore, we expected to see different responses to salinity stress between species, as well as between the glycophytic and halophytic tolerance classes.

Seeds for the glycophyte experiment were sourced from the same sources as in de la Reguera [57], with the exception of *P. maculosa*, which was collected from an urban, roadside population in northwest Washington, D.C., and *P. punctata*, which was collected from the understory of a maritime forest in The Nature Conservancy’s Brownsville Preserve in Nassawadox, Virginia, USA. Seeds for the halophyte experiment were provided by the USDA-NRCS Cape May Plant Materials Center, with the exception of *P. australis*, which was collected from the forest edge of Moneystump Swamp in Blackwater National Wildlife Refuge.

### 4.2. Experimental Design

Two salt stress exposure experiments were conducted in the Harlan Greenhouse in George Washington University’s Science and Engineering Hall. The first experiment, run from 15 March to 9 May 2021, tested the five glycophytes, and also included the moderately salt-tolerant *P. virgatum*. The second greenhouse experiment was run from 15 May to 16 July 2021 and tested the five halophytes. *P. virgatum* was included in both experiments to bridge results.

Salinity duration was controlled by the amount of time plants were subjected to salt water versus fresh water. Salinity intensity was controlled by the salt concentration of the saltwater. We exposed species to a range of salinity intensities appropriate to their salt stress tolerance class (maximum 6 PPT for the glycophyte experiment; maximum 24 PPT for the halophyte experiment). In the glycophyte experiment, intensity treatments were 0 (freshwater control), 1, 2, 4, and 6 PPT, and duration treatments were 0, 5, 10, 20, and 30 days of salinity exposure per month (DPM) (Figure 4a). In the halophyte experiment, intensity treatments were 0 (freshwater control), 6, 12, and 24 PPT, and duration treatments were 0, 5, 15, and 30 DPM (Figure 4b). Therefore, the glycophyte experiment included 17 combinations of salinity intensity and duration, while the halophyte experiment included 10 combinations (Figure 4). Saltwater was made by mixing Instant Ocean with deionized water in large containers to reach the target concentration. Plants were bottom-watered, and water was changed every 3 days to prevent salinity buildup in soils due to evaporation. During the water change for the halophyte experiment, soil was gently flushed with freshwater for 15 s to prevent the buildup of salinity in the soil, which proved a challenge during the glycophyte experiment (Figure A1 and Figure A2). In both experiments, trays were rotated among greenhouse benches at each watering to prevent greenhouse position effects. A handheld salinity meter (Pocket Pro+ Multi 1) was used to check the source water salinity of each treatment. The glycophyte experiment ran for 60 days, or two 30-day cycles, and the halophyte experiment ran for 90 days, or three 30-day cycles.

All plants used in this study were germinated from seed in George Washington University’s Harlan Greenhouse. To stimulate germination, seeds were spread on top of a wet potting mix (2:1 of potting mix:sand) and placed into a warm and humid germination chamber. Seeds for the halophyte experiment were treated with fungicide prior to germination. Trays of germinated seedlings were placed on heated mats until developing true leaves, at which point the plants were individually repotted and tagged with a unique identifier to facilitate tracking throughout the experiment.

The glycophytes were repotted into 10 × 10 × 10 cm cubic pots and placed into flat trays (Figure A3). Three individuals of each of the six glycophyte species were placed into each tray. In the halophyte experiment, seedlings were repotted into conical pots, and placed in a plant stand set atop six cylindrical containers (Figure A4). Each cylindrical container held five cones with one individual of each of the five halophytic species. All plants within a tray or container received the same water, and therefore the same stress treatments. The tray or container served as an experimental replicate and was randomly assigned to a combination of salt intensity and duration. Plants were haphazardly assigned a treatment combination. In each experiment, there were ten replicates of each treatment combination.

### 4.3. Response Variables—Survival, Biomass, and Chlorophyll

We screened for salt tolerance by measuring survival, biomass, and chlorophyll response to increasing salinity. We monitored survival every three days. Plants with no green aboveground tissue were scored as dead and removed from the experiment. At the end of the experiment, plants were removed from their pots, and gently agitated in water to remove the soil from the roots. Biomass was separated into aboveground live, aboveground dead, and belowground components. All biomass was dried at 60 ° for at least 72 h before being weighed. Chlorophyll measurements were made in the halophyte experiment only, in the week prior to harvest, using a chlorophyll meter (model CCM-200 plus, Opti-Sciences). We made three measurements of chlorophyll concentrations in the third fully expanded leaf, or in the second if the plant had only two leaves.

### 4.4. Statistical Analysis

Outliers were removed from the dataset via visual analysis prior to data transformations and model analysis. Plant mortality was coded as a binary variable (1 indicating dead and 0 indicating alive). To facilitate comparisons across species that varied in size and growth rate, total biomass and aboveground biomass were transformed to a standard normal distribution using the following equation:B_y_ = x − mean(x)/sd(x),(1)
where x is biomass of species y.

For each experiment, we ran a series of models that included intensity (PPT), duration (DPM), and their interaction, or only the salt stress exposure index (SSE), to identify the metric that best explained patterns of plant fitness and survival in response to experimental salinity stress in greenhouse manipulations. SSE was calculated by multiplying the PPT treatment by the DPM treatment to generate a single linear scale of salinity stress. Generalized linear mixed-models were built using the glm() function in R. In addition to these predictor variables, all models included species as a fixed factor to allow patterns to vary between species, and the tray as a random factor with a zero intercept.

In order to determine the metric(s) of salinity that best explained variation in survival and ecophysiology, we performed model selection based on the model with the lowest AIC value as the model of best fit for each experiment. We selected models for each experiment independently, as we expected glycophyte and halophyte groups to exhibit different responses to salinity stress. After selecting the model of best fit for the tolerance class, we ran the model for each of the species to identify the species-specific significance of predictor coefficients.

Standardized total biomass and standardized aboveground biomass met assumptions of normality and homogeneity of variance by definition; however, chlorophyll data were log-transformed prior to analysis to meet these assumptions. Total biomass was tested for assumptions of normality and homogeneity of variance prior to linear regression or ANOVA. Where ANOVA indicated a significant effect, pairwise comparisons were made using Tukey’s Highly Significant Difference tests.

Lastly, simple linear regression models were used to predict total biomass as a function of PPT within the highest duration treatment, 30 DPM, as a measure of each species’ sensitivity to salinity. A more negative regression coefficient indicated greater salinity sensitivity. Results were compared to species’ salt tolerances in the literature.

## 5. Conclusions

Our study investigated the independent and interactive effects of salinity intensity and duration on survival and ecophysiology for a selection of glycophytic and halophytic plant species. The study was limited in scale in that we tested only 10 species, 5 of each tolerance class, but the selected species encompassed a wide range of tolerance and a diversity of adaptations to salinity stress. Experiments were of limited duration. Despite these limitations, we identified that salinity intensity best explained explain patterns in plant survival in glycophytes, while the interaction between intensity and duration best explained variations in survival in halophytes, and biomass production for both glycophytes and halophytes. Combining duration and intensity into a single variable (the SSE index) oversimplified the effect of salinity stress on plant fitness, as we found non-linear effects and complex interactive effects of salinity intensity and duration on plant species’ responses. Most species experienced a cost in biomass production of growing in higher salinity that intensified with increasing stress duration. However, in chlorophyll production, some halophytes exhibited a positive, compensatory response to salinity, in which chlorophyll production increased with salinity intensity, and this effect was more pronounced at higher durations. Understanding interspecific variation in tolerance to salinity intensity and duration can identify management recommendations for agriculture, as well as help to predict shifts in composition, such as which sensitive species will be filtered from coastal plant communities or outcompeted by more salt-tolerant species.

## Figures and Tables

**Figure 1 plants-12-02522-f001:**
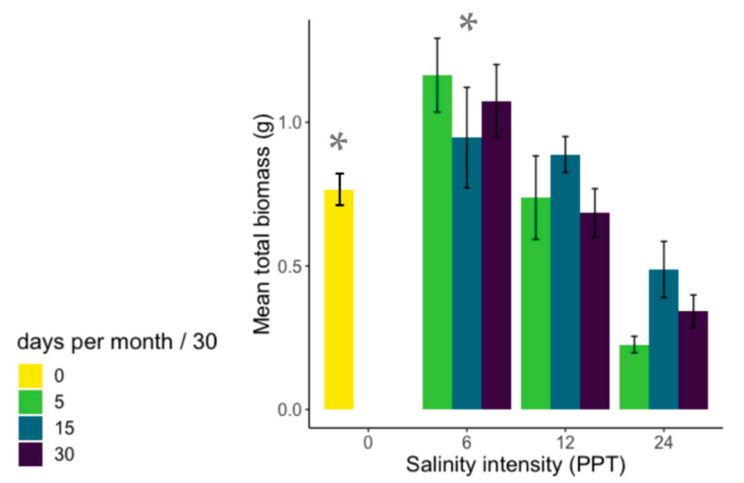
Mean total biomass (g) of *K. virginica* in each treatment combination of intensity and duration. *K. virginica* illustrates an example of non-linear effects of salinity; total biomass was higher in the 6 PPT treatment than 0 PPT (asterisks denote the significant difference between these two intensities in post hoc tests). PPT = parts per thousand.

**Figure 2 plants-12-02522-f002:**
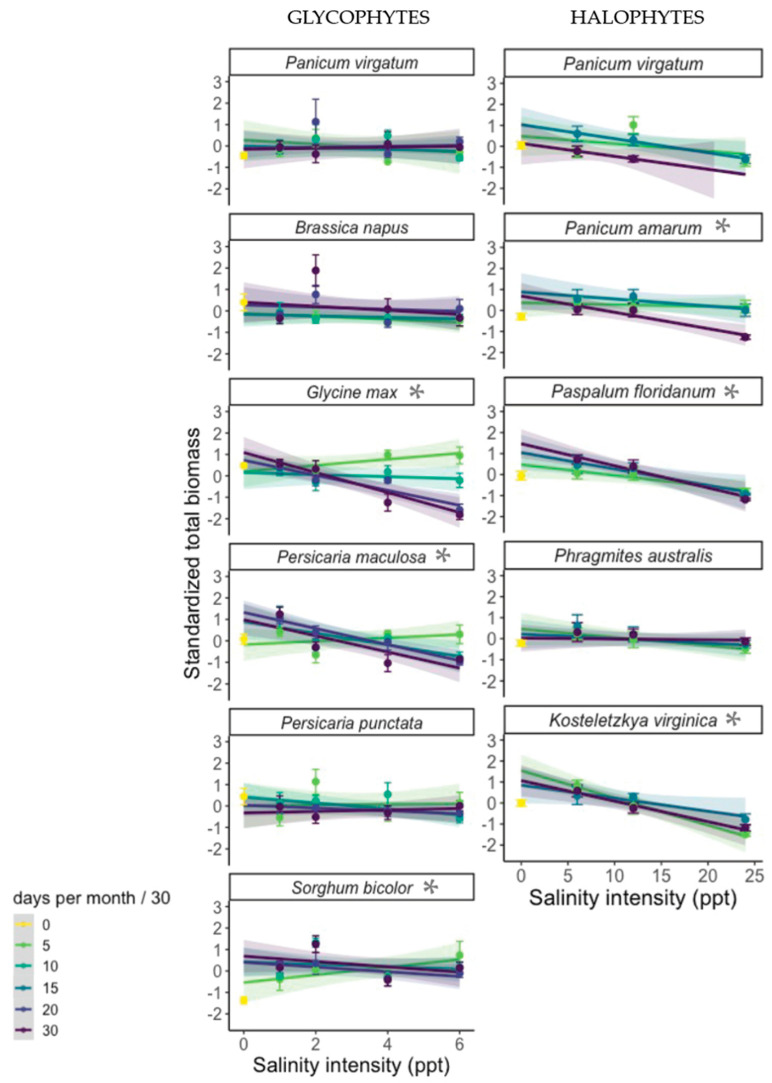
Standardized total biomass produced by the 10 tested species (glycophytes, left and halophytes, right) by salinity intensity (PPT) treatments. Colors distinguish salinity duration treatments. The yellow point indicates the control treatment, which received freshwater for the duration of each experiment. Where points are missing, all individuals in the treatment died prior to harvest. Shading around model curves indicates 95% confidence intervals. Asterisks by species’ names denote a significant interactive effect between intensity and duration.

**Figure 3 plants-12-02522-f003:**
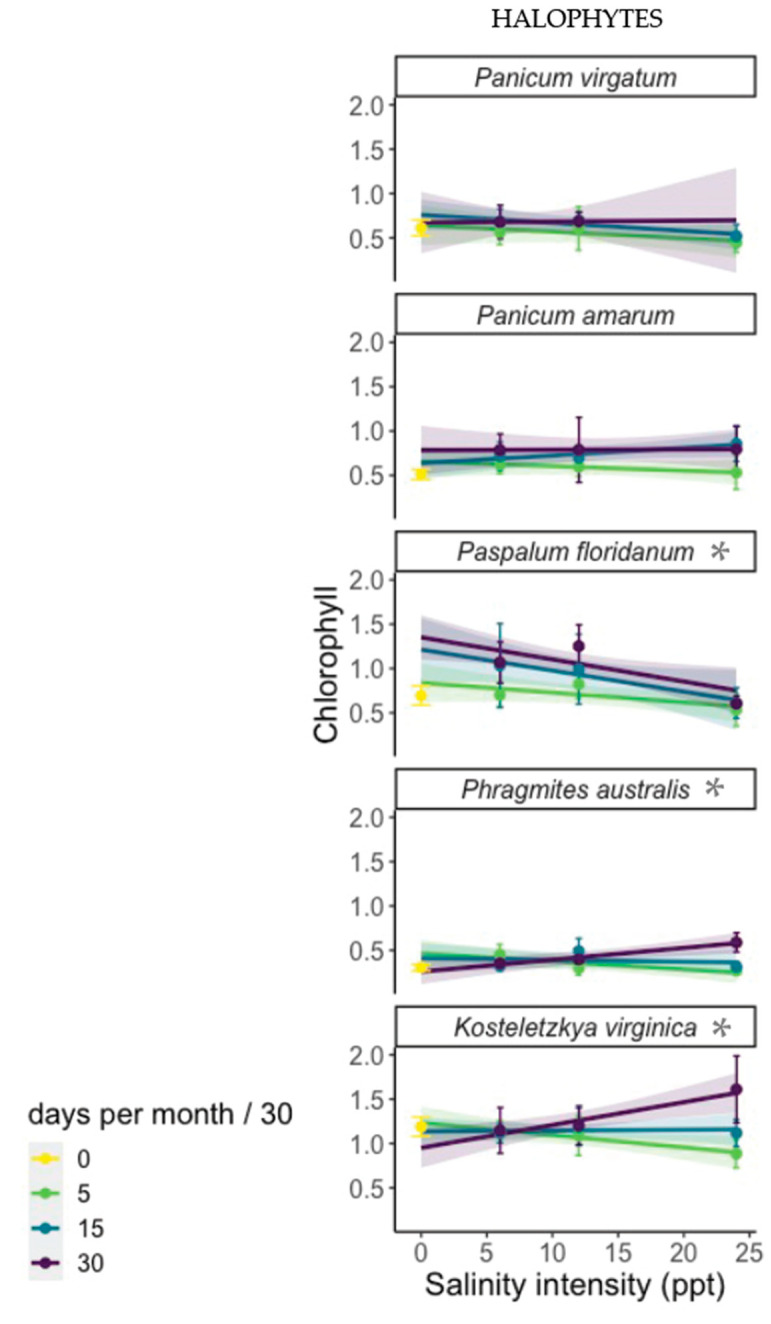
Log transformed chlorophyll by salinity intensity (PPT), grouped and colored by salinity duration. Asterisks denote a significant interactive effect between intensity and duration on chlorophyll.

**Figure 4 plants-12-02522-f004:**
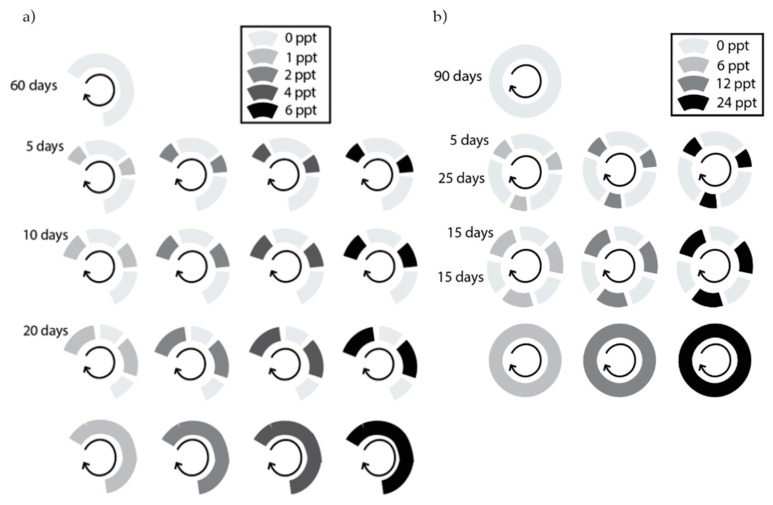
Diagram of the: (**a**) glycophyte experiment; and (**b**) halophyte experiment. Each circle represents the salinity exposure of an experimental unit throughout the experiment. Grayscale reflects salinity intensity treatments in: (**a**) 0 PPT (control), 1, 2, 4, and 6 PPT; and (**b**) 0 PPT (control), 6, 12, and 24 PPT; salinity duration treatments are shown as arcs within the circle, as: (**a**) 0 (control), 5, 15, and 30 days per month; and (**b**) 0 (control), 5, 10, 20, and 30 days per month. Experiments concluded after: (**a**) 90 days; and (**b**) 60 days.

**Table 2 plants-12-02522-t002:** Model selection using AIC (Akaike information criterion) values for all models. R^2^m = R^2^ marginal. DPM = days per month, i.e., duration. PPT = parts per thousand, i.e., intensity. SSE = salt stress exposure, i.e., the product of duration and intensity. The best performing model in terms of the lowest AIC is highlighted in red for each response variable within each experiment.

Experiment	Response Variable	Fixed and Random Predictor Variables	AIC	R^2^m
halophytes	survival	species * DPM * PPT + (0 + PPT | tray) + (0 + DPM | tray)	** 335.07 **	0.951
halophytes	survival	species * SSE + (0 + SSE | tray)	343.8	0.968
halophytes	survival	species * PPT + (0 + PPT | tray)	340.79	0.958
halophytes	survival	species * DPM + (0 + DPM | tray)	405.04	0.961
glycophytes	survival	species * DPM * PPT + (0 + PPT | tray) + (0 + DPM | tray)	238.51	0.961
glycophytes	survival	species * SSE + (0 + SSE | tray)	224.89	0.962
glycophytes	survival	species * PPT + (0 + PPT | tray)	** 219.45 **	0.962
glycophytes	survival	species * DPM + (0 + DPM | tray)	225.26	0.962
halophytes	standardized total biomass	species * DPM * PPT + (0 + PPT | tray) + (0 + DPM | tray)	** 1409.7 **	0.123
halophytes	standardized total biomass	species * SSE + (0 + SSE | tray)	1439.6	0.036
halophytes	standardized total biomass	species * PPT + (0 + PPT | tray)	1436	0.043
halophytes	standardized total biomass	species * DPM + (0 + DPM | tray)	1453.1	0.011
glycophytes	standardized total biomass	species * DPM * PPT + (0 + PPT | tray) + (0 + DPM | tray)	** 1559.2 **	0.185
glycophytes	standardized total biomass	species * SSE + (0 + SSE | tray)	1588.1	0.111
glycophytes	standardized total biomass	species * PPT + (0 + PPT | tray)	1617.6	0.066
glycophytes	standardized total biomass	species * DPM + (0 + DPM | tray)	1623.5	0.056
halophytes	standardized aboveground biomass	species * DPM * PPT + (0 + PPT | tray) + (0 + DPM | tray)	** 1424.2 **	0.098
halophytes	standardized aboveground biomass	species * SSE + (0 + SSE | tray)	1455.4	0.006
halophytes	standardized aboveground biomass	species * PPT + (0 + PPT | tray)	1453.8	0.009
halophytes	standardized aboveground biomass	species * DPM + (0 + DPM | tray)	1442.4	0.031
glycophytes	standardized aboveground biomass	species * DPM * PPT + (0 + PPT | tray) + (0 + DPM | tray)	** 1572.7 **	0.173
glycophytes	standardized aboveground biomass	species * SSE + (0 + SSE | tray)	1602.5	0.098
glycophytes	standardized aboveground biomass	species * PPT + (0 + PPT | tray)	1621.2	0.069
glycophytes	standardized aboveground biomass	species * DPM + (0 + DPM | tray)	1632.7	0.050
halophytes	log chlorophyll	species * DPM * PPT + (0 + PPT | tray) + (0 + DPM | tray)	** −15.57 **	0.596
halophytes	log chlorophyll	species * SSE + (0 + SSE | tray)	46.748	0.530
halophytes	log chlorophyll	species * PPT + (0 + PPT | tray)	67.648	0.511
halophytes	log chlorophyll	species * DPM + (0 + DPM | tray)	13.062	0.561

**Table 3 plants-12-02522-t003:** Percentage survival by intensity and duration in the halophyte experiment. Darker brownindicates lower mean survival; 100% survival indicates no mortality. DPM = days per month, i.e., duration. PPT = parts per thousand, i.e., intensity.

Species	Duration (DPM)	0 PPT	6 PPT	12 PPT	24 PPT
** *Panicum virgatum* **	**0**	**100%**			
** *Panicum virgatum* **	**5**		**100%**	**90%**	**80%**
** *Panicum virgatum* **	**15**		**100%**	**80%**	**50%**
** *Panicum virgatum* **	**30**		**100%**	**80%**	**0%**
** *Panicum amarum* **	**0**	**100%**			
** *Panicum amarum* **	**5**		**100%**	**100%**	**100%**
** *Panicum amarum* **	**15**		**100%**	**100%**	**100%**
** *Panicum amarum* **	**30**		**100%**	**100%**	**90%**
** *Paspalum floridanum* **	**0**	**96%**			
** *Paspalum floridanum* **	**5**		**100%**	**89%**	**89%**
** *Paspalum floridanum* **	**15**		**100%**	**89%**	**56%**
** *Paspalum floridanum* **	**30**		**100%**	**100%**	**44%**
** *Phragmites australis* **	**0**	**93%**			
** *Phragmites australis* **	**5**		**90%**	**90%**	**90%**
** *Phragmites australis* **	**15**		**90%**	**100%**	**70%**
** *Phragmites australis* **	**30**		**90%**	**70%**	**70%**
** *Kosteletzkya virginica* **	**0**	**100%**			
** *Kosteletzkya virginica* **	**5**		**91%**	**90%**	**50%**
** *Kosteletzkya virginica* **	**15**		**100%**	**90%**	**40%**
** *Kosteletzkya virginica* **	**30**		**80%**	**90%**	**40%**

**Table 4 plants-12-02522-t004:** Percentage survival by intensity and duration in the glycophyte experiment. Darker brown indicates lower mean survival; 100% survival indicates no mortality. DPM = days per month, i.e., duration. PPT = parts per thousand, i.e., intensity.

Species	Duration (DPM)	0 PPT	1 PPT	2 PPT	4 PPT	6 PPT
** *Brassica napus* **	**0**	**92%**				
** *Brassica napus* **	**5**		**100%**	**100%**	**100%**	**67%**
** *Brassica napus* **	**10**		**100%**	**100%**	**100%**	**100%**
** *Brassica napus* **	**20**		**100%**	**100%**	**80%**	**100%**
** *Brassica napus* **	**30**		**100%**	**100%**	**100%**	**100%**
** *Glycine max* **	**0**	**100%**				
** *Glycine max* **	**5**		**100%**	**100%**	**100%**	**80%**
** *Glycine max* **	**10**		**67%**	**100%**	**100%**	**100%**
** *Glycine max* **	**20**		**100%**	**100%**	**100%**	**83%**
** *Glycine max* **	**30**		**100%**	**100%**	**100%**	**80%**
** *Panicum virgatum* **	**0**	**91%**				
** *Panicum virgatum* **	**5**		**50%**	**100%**	**17%**	**50%**
** *Panicum virgatum* **	**10**		**83%**	**83%**	**67%**	**67%**
** *Panicum virgatum* **	**20**		**83%**	**100%**	**50%**	**67%**
** *Panicum virgatum* **	**30**		**83%**	**50%**	**67%**	**67%**
** *Persicaria maculosa* **	**0**	**100%**				
** *Persicaria maculosa* **	**5**		**100%**	**100%**	**100%**	**100%**
** *Persicaria maculosa* **	**10**		**100%**	**100%**	**100%**	**100%**
** *Persicaria maculosa* **	**20**		**100%**	**100%**	**100%**	**100%**
** *Persicaria maculosa* **	**30**		**100%**	**100%**	**100%**	**100%**
** *Persicaria punctata* **	**0**	**100%**				
** *Persicaria punctata* **	**5**		**100%**	**100%**	**100%**	**100%**
** *Persicaria punctata* **	**10**		**100%**	**100%**	**100%**	**100%**
** *Persicaria punctata* **	**20**		**100%**	**100%**	**100%**	**100%**
** *Persicaria punctata* **	**30**		**100%**	**100%**	**100%**	**100%**
** *Sorghum bicolor* **	**0**	**100%**				
** *Sorghum bicolor* **	**5**		**100%**	**100%**	**100%**	**100%**
** *Sorghum bicolor* **	**10**		**100%**	**100%**	**100%**	**100%**
** *Sorghum bicolor* **	**20**		**100%**	**100%**	**100%**	**100%**
** *Sorghum bicolor* **	**30**		**100%**	**100%**	**100%**	**100%**

**Table 5 plants-12-02522-t005:** Model predictor significance in species-specific models of total biomass. Columns describe the data transformation used to meet assumptions of normality and homogeneity of variance, and values are coefficient values, with p-values in parentheses. *p*-values are bolded to reflect significance of *p* < 0.05, and asterisks highlight significant *p*-values for the interaction between PPT and DPM with * *p* < 0.05, ** *p* < 0.01, and *** *p* < 0.001 DPM = days per month, i.e., duration. PPT = parts per thousand, i.e., intensity.

Species	Data Transformation	PPT	DPM	PPT*DPM
*Brassica napus*	square root	−0.038 (*p* = 0.137)	0.053 (*p* = 0.276)	0.00009 (*p* = 0.946)
*Glycine max*	standard normal	0.132 **(*p =* 0.029)**	0.021 (*p* = 0.081)	−0.022 **(*p* < 0.0001 ***)**
*Kosteletzkya virginica*	standard normal	−0.030 (*p* = 0.101)	0.042 **(*p =* 0.004)**	−0.003 **(*p =* 0.0252 *)**
*Panicum amarum*	standard normal	0.043 **(*p =* 0.004)**	0.039 **(*p =* 0.002)**	−0.004 **(*p<* 0.0001 *****)**
*Panicum virgatum (halophyte)*	log	−0.009 (*p* = 0.539)	0.014 (*p* = 0.322)	−0.002 (*p* = 0.210)
*Panicum virgatum (glycophyte)*	log	0.051 (*p* = 0.549)	0.014 (*p* = 0.367)	−0.003 (*p* = 0.603)
*Paspalum floridanum*	log	−0.013 (*p* = 0.413)	0.054 **(*p =* 0.0001)**	−0.003 **(*p =* 0.0092 **)**
*Persicaria maculosa*	standard normal	0.020 (*p* = 0.756)	0.041 **(*p =* 0.002)**	−0.016 **(*p <* 0.0001 ***)**
*Persicaria punctata*	standard normal	−0.076 (*p* = 0.308)	−0.022 (*p* = 0.137)	0.003 (*p* = 0.563)
*Phragmites australis*	log	−0.010 (*p* = 0.467)	0.021 (*p* = 0.091)	−0.0003 (*p* = 0.7413)
*Sorghum bicolor*	standard normal	0.281 **(*p <* 0.0001)**	0.068 **(*p <* 0.0001)**	−0.017 **(*p <* 0.0001 ***)**

**Table 6 plants-12-02522-t006:** Model predictor significance in species-specific models of chlorophyll concentration. Chlorophyll data were log-transformed to meet assumptions of normality and homogeneity of variance, and values are coefficient values, with *p*-values in parentheses. *p*-values are bolded to reflect significance of *p* < 0.05, and asterisks highlight significant *p*-values for the interaction between PPT and DPM with * *p* < 0.05, and *** *p* < 0.001 DPM = days per month, i.e., duration. PPT = parts per thousand, i.e., intensity.

Species	PPT	DPM	PPT*DPM
*Kosteletzkya virginica*	−0.016 **(*p =* 0.0002)**	−0.007 **(*p =* 0.017)**	0.001 **(*p <* 0.0001 ***)**
*Panicum amarum*	0.002 (*p* = 0.496)	0.008 **(*p =* 0.013)**	0.000008 (*p* = 0.975)
*Panicum virgatum*	−0.007 (*p* = 0.131)	0.004 (*p* = 0.363)	0.0001 (*p* = 0.834)
*Paspalum floridanum*	−0.006 (*p* = 0.289)	0.024 **(*p <* 0.0001)**	−0.0008 **(*p =* 0.045 *)**
*Phragmites australis*	−0.004 (*p* = 0.164)	−0.0003 (*p* = 0.911)	0.0004518 **(*p =* 0.023 *)**

**Table 7 plants-12-02522-t007:** The relative salt sensitivity of each species is reflected in the coefficient value of a linear regression of the effect of salinity intensity (PPT, part per thousand) on plants within the 30 DPM (days per month) duration treatment, as a loss (i.e., negative values) of potential biomass in grams per unit increase in PPT of salinity. Only data from the halophyte experiment was used to calculate the sensitivity score for *P. virgatum*, as this species is classified as a higher tolerance species.

Species	Tolerance Class	Salt Sensitivity Score (g/PPT)
*Persicaria punctata*	lower	0.042
*Phragmites australis*	higher	−0.007
*Panicum amarum*	higher	−0.022
*Panicum virgatum*	higher	−0.030
*Kosteletzkya virginica*	higher	−0.040
*Paspalum floridanum*	higher	−0.059
*Brassica napus*	lower	−0.092
*Sorghum bicolor*	lower	−0.131
*Persicaria maculosa*	lower	−0.530
*Glycine max*	lower	−0.599

## Data Availability

Data from this study are available on the Gedan Lab GitHub site, https://github.com/gedanlab/salinity-stress-intensity-and-duration.git (accessed on 29 June 2023).

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
