# Peer review of "Distinguishing the Effects of Stress Intensity and Stress Duration in Plant Responses to Salinity"

_plants, 2023, doi:10.3390/plants12132522_

Round 1

Reviewer 1 Report

Overall the paper is of interest and the manuscript is well written.  There are some questions related to the methods and a number of issues with the tables and figures that should be addressed before publication.

Title

  1. Looks fine

Abstract

  1. Well written and highlighting the most important information

Introduction

  1. Good justification for the study.
  2. Well written

Material and Methods

  1. It wasn’t clear how many replicates there were for each treatment combination?  
  2. Was the experiment set up as a randomized design?

Results and Discussion

  1. Please indicate differences among species and treatments on the figures if possible

Figures and tables

  1. Tables – In table two define all of the abbreviations used.
  2. In tables 3 and 4 do the colored boxes represent statistically significant decreases in survival?
  3. Figure 4 – indicate significant differences on the figure.  In addition, why is only the data for this species shown rather than all 10 species?
  4. The figures were very out of order, which made it confusing to connect the discussion of the data to the actual data.
  5. Figures 1 and 2 – do the lighter bands of color surrounding each line indicate a confidence interval, standard error, etc?
  6. Figure 3 should be relabeled to Figure 4 since it comes after the discussion of the data in figure 4.

Author Response

Title

  1. Looks fine

Abstract

  1. Well written and highlighting the most important information

Introduction

  1. Good justification for the study.
  2. Well written

Material and Methods

  1. It wasn’t clear how many replicates there were for each treatment combination?  

Added a sentence to the methods: “In each experiment, there were ten replicates of each treatment combination.”

  1. Was the experiment set up as a randomized design?

Added a sentence to the methods: “Plants of all species were haphazardly assigned a treatment combination.”

Results and Discussion

  1. Please indicate differences among species and treatments on the figures if possible

We added asterisks in figures where there was a significant interactive effect between intensity and duration, which was the most common type of effect. Beyond this, we felt that the figures would become too busy if additional statistical effects are shown, since they include many species and treatment combinations.

Figures and tables

  1. Tables – In table two define all of the abbreviations used.

Updated the Table two legend: “AIC (Akaike information criterion) values for all models including all species. R2m = R2 marginal. DPM = days per month, i.e. the duration treatment. PPT = parts per thousand, i.e. the intensity treatment. sse = salt stress exposure, i.e. the combination of duration and intensity. The best performing model in terms of lowest AIC is highlighted in red for each response variable.”

  1. In tables 3 and 4 do the colored boxes represent statistically significant decreases in survival?

Updated legend: “Table 3. Proportion survival by intensity and duration in the halophyte experiment. Darker red indicates lower mean survival; color does not indicate statistical significance. 100% means no mortality.”

  1. Figure 4 – indicate significant differences on the figure.  In addition, why is only the data for this species shown rather than all 10 species?

Updated legend: “Figure 1. Mean total biomass (g) of K. virginica of all plants in each intensity and duration treatment. K. virginica illustrates an example of non-linear effects of salinity; total biomass was higher in the 6 PPT treatment than 0 PPT. Asterisks denote significant differences between salinity intensity treatments in mean total biomass, summarized across days per month."

We show only K. virginica in Fig. 4, rather than all 10 species, because total biomass information was somewhat repetitive of the standardized total biomass data from Figure 1, and we felt that showing a single species better highlighted the issue of the non-linear response of halophytes. We can include all ten species as a panel figure in the appendix, if desired.

  1. The figures were very out of order, which made it confusing to connect the discussion of the data to the actual data.

Corrected the numbering of the figures in the manuscript.

  1. Figures 1 and 2 – do the lighter bands of color surrounding each line indicate a confidence interval, standard error, etc?

Updated legend: “95% confidence intervals are indicated by shading around model curves.”

  1. Figure 3 should be relabeled to Figure 4 since it comes after the discussion of the data in figure 4.

Fixed

Reviewer 2 Report

The This study investigated the effects of presses and pulses of salinity stress on five glycophytic and five halophytic species to determine whether salinity intensity, duration, or their interaction best explain patterns in survival and performance. The results of this study may contribute to reveal how species' responses vary in magnitude and by tolerance class. However, there are some concerns that the authors should address before it can be considered for publication.

(1) The introduction is general and unspecific. The gaps and motivations of this study need to be further introduced and discussed.

(2) The introduction contains many previous studies by the authors, and it is recommended to streamline and summarize more of the previous research results.

(3) I suggest the authors add the research significance of this article in the last paragraph of the introduction.

(4) More mechanism explanations should be added to further explain the different of salinity stress effects.

(5) In order to further highlight the innovation of this article, it is better to compare the results of this study with other studies.

(6) A paragraph of limitation discussion should be added to clarify the limitation or uncertainty of the design and methods of the experiment in this current study.

Author Response

(1) The introduction is general and unspecific. The gaps and motivations of this study need to be further introduced and discussed.

Added: “This is the impetus for our study, which seeks to investigate, on a species-level, how plants with different salinity adaptations will respond to elevated stress along multiple axes of exposure.”

(2) The introduction contains many previous studies by the authors, and it is recommended to streamline and summarize more of the previous research results.

We citations referencing new literature from outside of the Gedan Lab and our direct collaborators.

(3) I suggest the authors add the research significance of this article in the last paragraph of the introduction.

Added: The research significance of this study is to identify the metric of salinity that best explains plant physiology and mortality, detect whether the best metric is different between glycophytes and halophytes, and, particularly amongst the crops, identify which species might perform best under elevated salinity conditions. This research will aid in developing adaptation strategies as coastal communities experience elevated salinity due to SLR and SWI. As salinity is a complex environmental variable and our study includes two groups with distinct adaptation strategies (glycophytes and halophytes), identifying the metric(s) that best explain impacts on plant survival and ecophysiology will aid in planning adaptation and restoration projects in coastal communities.

(4) More mechanism explanations should be added to further explain the different of salinity stress effects.

Added: “This is likely due to this species’ salinity adaptations [Table 1]; previous study has demonstrated efficient sodium exclusion [43] along with the ability to adjust osmotic solutes in their leaves [44], allowing P. australis to survive and grow under high salinity intensity even over long periods.”

Added: “Despite previous study demonstrating the ability of this species to contain salt ions in their roots and stem [49], and the variety of G. max used in our study being a chloride-excluding variety, it did not perform well in this experiment or in field trials with the same variety in salt-stressed fields [50].”

Added: “Based on our findings, S. bicolor would be a better choice of crop for salt-affected coastal areas than G. max, even without pre-treatment to salinity [47]. This supports the use of S. bicolor in salt-affected areas, as encouraged by previous studies that have identified this crop species of interest for areas impacted by coastal salinization [51].”

(5) In order to further highlight the innovation of this article, it is better to compare the results of this study with other studies.

Added: “The development of indices that can more simply describe salinity stress effects on plants and can generalize these effects to large groups of species, such as glycophytes or halophytes, would fulfill a research and communication need.”

Added: “While Li and Pennings investigated mesocosm communities, our study focused on species-specific effects of salinity stress on survival; we aimed to investigate species when individually exposed to elevated salinity conditions in order to anticipate and describe how communities will develop in the face of salinity. Based on our findings, we expect the less tolerant glycophytes, such as P. maculosa, to be the first ones filtered from the community, while more tolerant species will persist as coastal salinity regimes shift.”

Added: “Our experiment lacked any sort of pre-treatment, choosing instead to mimic real-world conditions where plants are exposed to novel salinity conditions without the opportunity for acclimation in S. bicolor or any other species.”

(6) A paragraph of limitation discussion should be added to clarify the limitation or uncertainty of the design and methods of the experiment in this current study.

Added: “Our study investigated the independent and interactive effects of salinity intensity and duration on survival and ecophysiology for a selection of glycophyte and halophyte coastal plant species. The study was limited in scale in that we were only able to include 10 species, five of each classification, though these species encompassed a wide range of halophytic adaptations. We were also limited in the time duration of this experiment; as such, our conclusions are limited to initial survival and ecophysiology, and should be interpreted as such.”

Reviewer 3 Report

In the manuscript, entitled "Distinguishing the effects of stress intensity and stress duration in plant responses to salinity", the authors concluded that in case of glycophytes the salinity intensity is the best to explain plant survival pattern. While in case of halophytes the interaction between salinity intensity and duration is the best. The manuscript is written in good manners and order. The methods are described in very much details. The used methods are adequate to investigate their hypothesis. The results are clearly written, the discussion part is easily leading the reader toward the conclusion. The findings of this manuscript are very interesting and I suggest it's publication.

Author Response

We humbly thank Reviewer 3 for these kind words.